# Trends and Dynamics in the First Four Years of Operation of the First Human Milk Bank in Vietnam

**DOI:** 10.3390/nu13041107

**Published:** 2021-03-28

**Authors:** Hoang Thi Tran, Tuan Thanh Nguyen, Debbie Barnett, Gillian Weaver, Oanh Thi Xuan Nguyen, Quang Van Ngo, Huong Thi Thanh Le, Le Thi Huynh, Chung Thi Do, Roger Mathisen

**Affiliations:** 1Neonatal Unit and Human Milk Bank, Da Nang Hospital for Women and Children, Da Nang 50506, Vietnam; xuanoanh0901@gmail.com (O.T.X.N.); lehuongnhqn@gmail.com (H.T.T.L.); bongdiendien26@gmail.com (L.T.H.); 2Department of Pediatrics, School of Medicine and Pharmacy, Da Nang University, Da Nang 50206, Vietnam; 3Alive & Thrive Southeast Asia, FHI 360, Hanoi 11022, Vietnam; tnguyen@fhi360.org (T.T.N.); chung.do@vnuk.edu.vn (C.T.D.); rmathisen@fhi360.org (R.M.); 4Milk Bank Scotland, Queen Elizabeth University Hospital, Glasgow G51 4TF, UK; Debbie.Barnett@ggc.scot.nhs.uk; 5International Milk Banking Specialist and Consultant, Human Milk Foundation, Harpenden AL5 2JQ, UK; gillian.weaver@yahoo.com; 6Department of Nutrition, Da Nang Center for Disease Control, Da Nang 50206, Vietnam; ngovanquang.dng@gmail.com

**Keywords:** human milk bank, COVID-19, donor milk, prelacteal feeding, breastfeeding, newborn, early essential newborn care (EENC)

## Abstract

Background: Since 1979, the World Health Organization (WHO) and the United Nations Children’s Fund (UNICEF) have recommended the use of pasteurized human milk from a human milk bank (HMB) to feed low birthweight (LBW) and preterm newborns as the ‘first alternative’ when mothers are unable to provide their own milk. However, they have not issued any guidelines for the safe establishment and operation of an HMB. This gap contributes to the demand for gathering experiences from HMB networks, especially those from lower-middle income countries. To fill this knowledge gap, this study examines the characteristics of donors, donation, pasteurization, and recipients during the first four years of operation in the first HMB in Vietnam. Methods: Data about the donors, donation, pasteurization, and recipients were extracted from the web-based electronic monitoring system of the HMB from 1 February 2017 to 31 January 2021. Results: In the first four years of operation there were 433 donors who donated 7642 L of milk (66% from the community) with an increased trend in the amount of donated milk, donation duration, and average amount of milk donated by a donor. Approximately 98% of the donated milk was pasteurized, and 82% passed both pre- and post-pasteurization tests. Although the pass rate tended to increase with time, a few dips occurred. Of 16,235 newborns who received pasteurized donor milk, two thirds were in the postnatal wards. The main reason for the prescription of pasteurized donor milk was insufficient mothers’ own milk in the first few days after birth. There was a decreased trend in the amount and duration of using pasteurized donor milk in both postnatal wards and the neonatal unit. Conclusions: The HMB has operated efficiently in the previous four years, even during the COVID-19 pandemic, to serve vulnerable newborns. Ongoing evidence-based adjustments helped to improve the operation to recruit suitable donors, to increase the access to and quality of raw donor milk, to improve the pasteurization process, and to meet the need of more newborns.

## 1. Introduction

Neonatal mortality contributes to more than 50% of total deaths among children under five years old in the Western Pacific Region including in Vietnam [1]. Breastfeeding is the most important intervention to reduce child mortality and morbidity, especially among sick or preterm newborns in the first 28 days of life. However, not all newborns have the same chance of receiving breastmilk from their own mothers because of the mothers’ death, absence, or inability to provide any or sufficient breastmilk because of certain diseases [2] or circumstances related to the childbirth or gestational age. To provide the best nutrition for these newborns, human donor milk is the alternative recommended by the World Health Organization (WHO) and the United Nations Children’s Fund (UNICEF) [3,4]. When quantities of pasteurized donor milk (PDM) are limited, the use by sick or preterm newborns should be prioritized [5]. In surplus, although there is no specific guidance, PDM is prescribed for healthy newborns with hypoglycemia and/or hyperbilirubinemia, excessive weight loss, delayed lactogenesis, small for gestational age, and maternal-newborn separation [6]. The use of PDM was associated with a 46% reduction in the incidence of necrotizing enterocolitis compared with infant formula (term or preterm) in preterm or low birthweight newborns [7], a 19% reduction in the odds of developing sepsis for every 10 mL per kg each day in the first 28 days of life in very low birthweight newborns compared to infant formula [8], a 22% lower incidence of bronchopulmonary dysplasia, and required almost 3 fewer days of ventilator support than preterm newborns who were supplemented with preterm formula [9].

The first human milk bank (HMB) in Vietnam was established in February 2017 as a result of a collaboration among Maternal and Child Health Department Vietnam Ministry of Health, Da Nang Health Department, Alive & Thrive and PATH, support from the Donor Milk Bank, Queen Elizabeth University Hospital, Scotland, and international experts on HMB [10]. The HMB is located in Da Nang Hospital for Women and Children (DNHWC), which is a referral hospital for maternal and child health in the Central region of Vietnam with more than 15,000 live births per year. The neonatal unit receives approximately 4000 admissions with a variety of conditions including medical and surgical challenges and prematurity [11]. We previously published the experience of establishing this HMB [10].

As of 31 January 2021, this HMB had received 7642 L of donor milk from 433 donors and administered PDM to 16,235 newborns over a four-year period. During this period there have been adaptations in practice to fulfil the demand of the HMB, donors, and recipients. Why were these changes required and what impact did they have? To fill in the knowledge gap, we conducted this study to examine characteristics of donors, donation, pasteurization, and recipients during the first four years of operation in the first HMB in Vietnam. This study further supports the increased demand for learning about experiences in the establishment and operation of HMB in East Asia and the Pacific region. This study also contributes to information needed to strengthen the HMB network in the region.

## 2. Methods

### 2.1. Data Sources and Study Variables

Data were extracted from a web-based electronic monitoring system used within the HMB [10] from 1 February 2017 to 31 January 2021. We included all records of donors, donation, pasteurization, and recipients from the monitoring system. The information from the donors included age, number of children, place of residence, education, occupation, place and mode of birth, the amount of donor milk and the duration of donation. This information was stratified by the place of donation (either the hospital or community) and time period (every six months or semester).

The information about pasteurization included the amount of donor milk pasteurized, and the amount that passed the pre- and post-pasteurization microbiology tests. The information was stratified by the place of donation (either the hospital or community) and time (every semester).

The information from the recipients included their general characteristics: modes of childbirth, very low birthweight or preterm births, and reasons for these birth outcomes. We also presented data on the volume of milk distributed, number of newborns who received PDM, average number of days using PDM, and median amount used. The information was stratified by the location of the child (postnatal ward vs. neonatal unit) and time (every semester).

### 2.2. Data Management and Analysis

Individual level data were extracted from the HMB electronic system [10]. Before data analysis, we excluded all identifiable information such as (1) name, telephone number, address (other than provinces) of the donors; and (2) name and date of birth of the newborns and their parents’ telephone number, address (other than province).

We performed descriptive analysis to present the indicators across three main stages of HMB operation: (1) collection of raw donor milk, (2) handling and processing of donor milk, and (3) usage of PDM. For each of the stages, we presented a summary table for the four-year estimates. Then, we presented some indicators by time and stratified by location of donors (e.g., from the hospital or community), amount of donor milk pasteurized (e.g., from the hospital or community), and recipients (e.g., neonatal unit or postnatal wards).

### 2.3. Ethical Considerations

Our research complies with the World Medical Association Declaration of Helsinki regarding the ethical conduct of research involving human subjects [12]. The monitoring data were collected based on standardized operational procedures and the secured online system developed based on learning from other HMB and through consultation with HMB experts and approved by the hospital and Da Nang Department of Health [10]. Written informed consent for the use of data for research and program improvement were obtained from all donors and recipients’ parents as a part of informed consent relating to the donation and use of donor milk [10].

## 3. Results

### 3.1. Characteristics of Human Milk Donors and Donation

During the four years of implementation (from February 2017 to January 2021), 433 mothers donated their breastmilk and 15 of them continued donating while this study analyzed the data. Donors had a mean age of 28.4 (standard deviation = 4.1) years; 69.8% resided in Da Nang City, 54.3% had an education level of college degree or higher, and 74.6% had a white-collar job (Table 1). Seventy nine percent of the donors gave birth at DNHWC, by cesarean section (57.0%) and with a preterm birth (43.0%) (Table 1). Around 50% of mothers started their donation in the hospital, whilst a similar proportion started donating after being discharged from hospitals. Key socio-demographic characteristics of donors such as age, profession, education, place of residence did not change much with time (data not shown).

Overall, the number of new donors decreased with time: from 92 people in the first six months to 23 in the last six months (Figure 1). The number of donors from the community increased from 32% in the first semester to around 50% to 65% in subsequent semesters (Figure 1).

The 433 donors donated 7642 L of milk of which 5063 L (66.3%) were from the community (Table 2). The amount of donor milk increased from 749 L in the first semester to around 1000 L in subsequent semesters but reduced to 742 L in the last semester (Figure 1). The amount of donor milk collected from donors in the hospital decreased from the first two semesters (431 and 443 L) with main drops in semesters 3, 4 and 8. In contrast, the amount of milk collected from donors from the community increased from the first semester (318 L) to the highest point in the fourth semester (964 L) and then decreased to about 640 L in semesters 5–7 and dropped to 517 in the last semester (Figure 1). The median duration of milk donation increased from about 20 days in the first two semesters to almost 90 days in semester 5, but then decreased and went down to about 50 days in the last semester. The median amount of donor milk per donor increased from 5 L in the first semester to more than 26 L in semester 7, but then decreased to about 18 L in the last semester (Figure 1).

### 3.2. Handling and Processing of Human Donor Milk

Of the 7642 L of milk donated, 7492 L (98.0%) were pasteurized (Table 2). The volume of donor milk pasteurized increased from 626 L in the first semester to about 1000 L in semesters 2–7 but dropped to 669 L in the last semester (Figure 1). About two thirds of the processed milk (5007 L) was from the community. The overall rate for passing both pre- and post-pasteurization tests was 81.7% and higher for donor milk obtained from the community (85.5%) than the hospital (73.7%). This rate improved from 60% in the first semester and 54% in the second semester to more than 80% in semesters 3–5 and more than 90% in semesters 7–8 (Figure 2). However, there was a reduction in the pass rate during the second and the fifth semesters from the community; and during the second and the seventh semesters from the hospital (Figure 2). The top three most common bacteria present in the milk pre- and post-pasteurization were *Bacillus*, *Staphylococcus aureus* and *Staphylococcus epidermidis*.

### 3.3. The Usage of Pasteurized Donor Milk

From February 2017 to January 2021, there were 16,235 newborns who received PDM, among those 5073 (31.2%) were from the neonatal unit, 11,162 (68.8%) were from postnatal wards (Table 3). Mothers without sufficient breastmilk in the first few days after birth was the main prescription reason for PDM in postnatal wards (95.3%) and neonatal unit (87.9%) (Table 3). The number of newborns who received PDM increased with time in the neonatal unit and postnatal wards (Figure 3) except for the last semester. However, the average amount of PDM and duration of PDM used decreased especially from the last semester in both neonatal unit and postnatal wards (Figure 3).

## 4. Discussions

### 4.1. Characteristics of Human Milk Donors

Over the four-year period, the total number of donors and proportion of donors with preterm births went down. The number of donors from the community increased over time while the number of donors from the hospital decreased. The majority of donors were having their first child, lived in Da Nang city, had a college degree or higher, had a white-collar job, and gave birth at DNHWC.

The number of donors decreased because after six months of operation we found that although there were many donors, their donation duration was short. Therefore, we changed our selection criteria to focus on donors who committed to donate for longer periods and in bigger amounts.

More than 80% of donors having given birth in DNHWC indicated the effectiveness of education and the promotion of breastfeeding and HMB in the hospital through direct consultation or public media. The donors who stayed in Da Nang city were preferred because of its convenience for the transportation of donor milk. There were also around 30% of donors from other provinces who gave birth in our hospital, mostly mothers of preterm newborns receiving care in our neonatal unit.

The mean age of donors in the HMB at DNHWC was 28.4 years, which was similar to that in HMB in Taiwan or China [13,14,15], higher than that in India (66% donors were under 25 years old) [16,17], while in the US the number of donors among two groups younger or older than 30 were similar [18]. While the majority of our donors have a college or undergraduate degree, in the study from India, the majority of donors had secondary school education or less [16]. More than 70% of our donors had a white-collar job. In Vietnam, employed women are given six months paid maternity leave [19], which enables them to breastfeed their children and donate surplus breastmilk to nearby HMB. Furthermore, this group might have more opportunities to access information and a higher commitment to follow the necessary hygiene recommendations for breastmilk expression and storage. Our finding is similar to a study in Brazil where women with higher education had twice the frequency of donation [20].

A high percentage of childbirth by cesarean section (up to 50%) was found in donors because the DNHWC is a referral hospital for high-risk pregnancy in the Da Nang city and surrounding area. Currently the cesarean section rate is around 55–60% [21]. Another special characteristic from Da Nang HMB was the high proportion of mothers with preterm births who donated breastmilk. This differed from HMBs in Taiwan and China, where the rate was less than 10% compared to more than 40% in our HMB. The reason for this is that many newborns in the preterm group were recipients of PDM, their mothers therefore understood the importance of donation and were more motivated to become donors [22]. At DNHWC, kangaroo mother care (KMC) is promoted for preterm and low birthweight newborns as early as possible even when the newborn is on respiratory support with continuous positive airway pressure (CPAP). When mothers stay in the neonatal unit with their newborns for KMC, they are supported to breastfeed and taught how to express and store their breastmilk. As they received the early effective intervention to protect and promote their milk supply, they are a good source of donations [2]. Many of the donors started donating when their newborns were one week old and donated for several weeks. These mothers could provide preterm breastmilk with higher calories and concentration of fat and protein for the preterm recipients [23,24]. Premature breastmilk is better for preterm newborns because it may contain higher levels of a number of important substances including epidermal growth factor that is essential for intestinal cell development; leptin that plays an important role in neonatal growth; glycosaminoglycans that prevent bacterial adherence to enterocytes; and exogenous antioxidants that prevent inflammation [25,26]. However, during the four-year operation the number of donors with preterm births decreased over time. Possible reasons include the workload in the neonatal unit led to a reduction in the active recruitment of donors. Secondly, recruitment criteria became stricter in enrolling mothers with a prolonged donation period. Thirdly, the number of donors from the community increased and compensated for the need for the hospital donors. However, mothers practicing KMC in the hospital should continue to be actively recruited as donors since they are easy to approach, the transportation of donor milk is convenient, and donor milk from preterm mothers is optimal for preterm newborns.

### 4.2. Human Donor Milk

Over the four-year period, the HMB received from under 800 L to more than 1000 L in each six-month period. The average amount per donor also tripled from under 5 L to more than 16 L with the average duration of donation from 22 days to nearly 40 days. This was achieved by providing education for potential donors and screening those who were able to commit to a longer donation period. This was cost effective in reducing the fees for the screening of donors (i.e., testing for human immunodeficiency virus (HIV), syphilis, hepatitis B and C once every six months). These longer-term donors were more likely to provide donor milk that passed microbiological tests. The average duration of donation in Korea was 3 months; in Taiwan, one donor donated 17 L, while in Norway this was 28 L [13,27]. A study of 3764 donors from an HMB in Northeastern United States (US) between 2011 and 2019 reported the median total volume per donor to be 11.4 L [28]. Our volume per donor has increased significantly with time and is comparable to Taiwan and the HMB in Northeastern US. The duration of donation in Da Nang was shorter because the hospital donors often stopped when their infant was discharged especially when they lived far from the hospital, or they did not have freezers at home suitable for milk storage. Our future work should focus on careful screening for suitable donors and donation duration, which can make the process more efficient.

During the coronavirus disease 2019 (COVID-19) pandemic in 2020, Da Nang city in Vietnam experienced a pandemic peak in August–September 2020 when Da Nang reported nearly 400 positive COVID-19 cases. During this time, the city was placed on social distancing measures and had limited transport. The number of new donors and the volume of donor milk in Da Nang decreased during the pandemic, which was similar to the situation in China, Poland, and India [29].

### 4.3. Microbiology of Human Donor Milk

The donor milk pre-pasteurization pass rate for microbiology tests increased over the four-year period. For the donor milk from the community, the pass rate rose from 85% in the first year to more than 95% in the fourth year. For the donor milk from the hospital, this rate increased from under 70% to more than 90%. The pre-pasteurization microbiology test’s aim is to check the hygiene quality and process of breastmilk expression and storage both in the hospital stations and donors’ homes. In the HMB, in order to improve the pass rate, we made several changes in donor education. The hygiene practice focused on improving the process for cleaning the containers and pump circuits used for expressing breast milk, sterilizing, and drying equipment properly. We also lent individual breast pumps to donors who did not have their own pumps instead of using shared hospital pumps as at the beginning of the HMB operation. We visited the donors’ houses to check the condition of the freezer and guided the donors to store donor milk in boxes, separated from their food.

For post-pasteurization tests, the donor milk from the community had a pass rate of under 70% in the first six months and increased to more than 90%. Similarly, the pass rate of the donor milk from the hospital increased from under 60% to more than 90%. Post-pasteurization microbiology results improved after changes in the process of milk expression practice, storage, close monitoring of the pasteurization cycle, and eliminating milk contamination. Three possible reasons associated with the positive results for microbiology tests after pasteurization include the pasteurization process, donor milk contaminated with *Bacillus* spores, and samples contaminated from the lab pathogens. *Bacillus* was sometimes present in our neonatal unit water in routine checks, and this possibly contaminated the donor milk via hand washing. We have subsequently focused on monitoring hand and equipment hygiene of donors. Pasteurization was completed following strict protocols and monitored closely. All problems identified have been addressed to ensure the process followed protocols. We also amended the protocol for laboratory testing by using an unopened full container, instead of taking a small sample from the container. This decreased the risk of cross contamination. Furthermore, at the laboratory, more infection control measures were initiated especially cleaning surfaces, ultraviolet sterilization of the laminar flow hood for culturing the samples before testing, appropriately handling samples, and using a negative control sample to eliminate false positives. Following these changes, the microbiology results improved significantly.

Overall, four-year results showed the pass rates pre- and post-pasteurization of 80% which is lower than that in Korea (87%) [30], China (95.6% pre-pasteurization) [15] and Taiwan (72% pre- and 99% post-pasteurization). Our results are comparable to that from a French study when non-compliance rate reached 25.9% of which 11.7% was after pasteurization and 17.7% before pasteurization [31]. The high discard rate in Da Nang HMB indicates that continuing quality improvement is important. 

### 4.4. Recipients

More than 16,235 newborns received PDM; 31% were from the neonatal unit and 69% were from the postnatal wards. Compared to HMB in Taiwan and Scotland, where they reported the provision of PDM was limited to very preterm or sick newborns in the neonatal unit [32], in Da Nang we provided PDM within three months of operation to all the newborns in need once we had sufficient donor milk. This was driven by parents of healthy newborns who had difficulties in breastfeeding asking for PDM for their newborns as an alternative to infant formula. The findings are consistent with those from a recent study in the US showing that 29% of Massachusetts birth hospitals and 43% of all birth hospitals served by the HMB in Northeastern US reported using PDM for healthy newborns, with the trend increasing [33]. There are concerns that HMBs may increase the dependence of mothers on PDM and reduce breastfeeding rates. However, evidence showed that PDM prevented the use of infant formula, increased breastfeeding and exclusive breastfeeding during the hospital stay and after the hospital discharge [34] and also prevented wet nursing which poses a risk to the newborns [35]. Positive findings were found in the US study with exclusive breastfeeding rates at the hospital discharge higher when PDM was used for healthy newborns (77% versus 56%, *p* = 0.02) [33]. This reinforces that the presence of an HMB is not just a place for collecting, storing, and providing PDM to mothers and newborns but can also promote breastfeeding by monitoring the number of PDM users, length of use and measures to support mothers to overcome breastfeeding difficulties [5,10].

The cesarean section rate was significantly higher among recipients with more than 60% in the neonatal unit and 70% in postnatal wards. Cesarean section is a well-known barrier for breastfeeding due to pain, stress, or anesthetics [36,37,38]. Further support for mothers with cesarean section is important. In addition, the breastfeeding consultation should start from prenatal visits and continue throughout the perinatal and postnatal periods.

The mean duration of use in the postnatal wards decreased from two days to one day and has been maintained at three days in the neonatal unit. The duration in Da Nang is much lower than that in Taiwan where 40% of recipients receive PDM for more than one month [13]. In Scotland, the mean duration was 14 days for preterm and 2 days for term newborns [32]. Da Nang is a Centre for Early Essential Newborn Care, and designated Center of Excellence for Breastfeeding, with high rates of early and prolonged skin-to-skin contact that enables mothers to be with their newborns for early and exclusive breastfeeding [21]. In addition, our KMC program for preterm and low birthweight newborns has run for more than ten years and mothers are encouraged to provide skin-to-skin contact as soon as the newborn is stable. Even during the COVID-19 pandemic, with strict infection prevention control measures, early skin-to-skin contact for all newborns and KMC for preterm and low birthweight newborns were continued [39]. Both early, prolonged skin-to-skin contact and KMC are important to stimulate mothers’ own milk, reduce the duration and amount of PDM use, speed up breastfeeding and for encouraging these mothers to become donors [2]. For other sick newborns, mothers are usually encouraged to take care of their newborns whenever they are stable, and the mothers are available. This measure ensures limited mother–newborn separation and promotes early breastfeeding for sick newborns.

During the COVID-19 pandemic in Da Nang, the number of recipients decreased as seen in the last six-month period because there was a decrease in referrals of high-risk pregnant women and newborns from surrounding provinces. In Da Nang there was only one woman with confirmed COVID-19 in labor in August 2020 [40]; this newborn received early skin-to-skin contact and was breastfed after birth. The newborn only required PDM for 36 h after cesarean section and the mother went on to achieve exclusive breastfeeding. Although there was a decrease in donors and donations during the COVID-19 pandemic, there was also a decrease in demand. Subsequently, the HMB did not face significant challenges during the COVID-19 pandemic.

To our knowledge, this study is among the few studies that has gathered experiences from the HMB networks, especially those from lower-middle income countries. The use of data gathered from secondary data collected from an online monitoring system helped to reduce the cost and reduced recall bias. The use of the online system with structured forms and pre-coded options as well as built-in verification functions help to reduce the workload of health workers and ensure the quality of the data. In addition, because the data from this monitoring system are regularly used to optimize the functionality of the HMB, ensure the compliance to standardized protocols and facilitate networking and information sharing among HMBs, the quality of the data is verified and improved.

## 5. Conclusions

After four years of operation, the first HMB in Vietnam has demonstrated the feasibility of building and maintaining an HMB with high standards in a low-middle income country. An HMB requires sustainable and ongoing support for long-term operational success and quality assurance. The application of evidence-based practice needs to continue to improve the HMB operation, the recruitment of suitable donors, to improve the quality of raw donor milk, provide better pasteurization processes and to meet the need of more infants, as well as supporting and improving breastfeeding.

## Figures and Tables

**Figure 1 nutrients-13-01107-f001:**
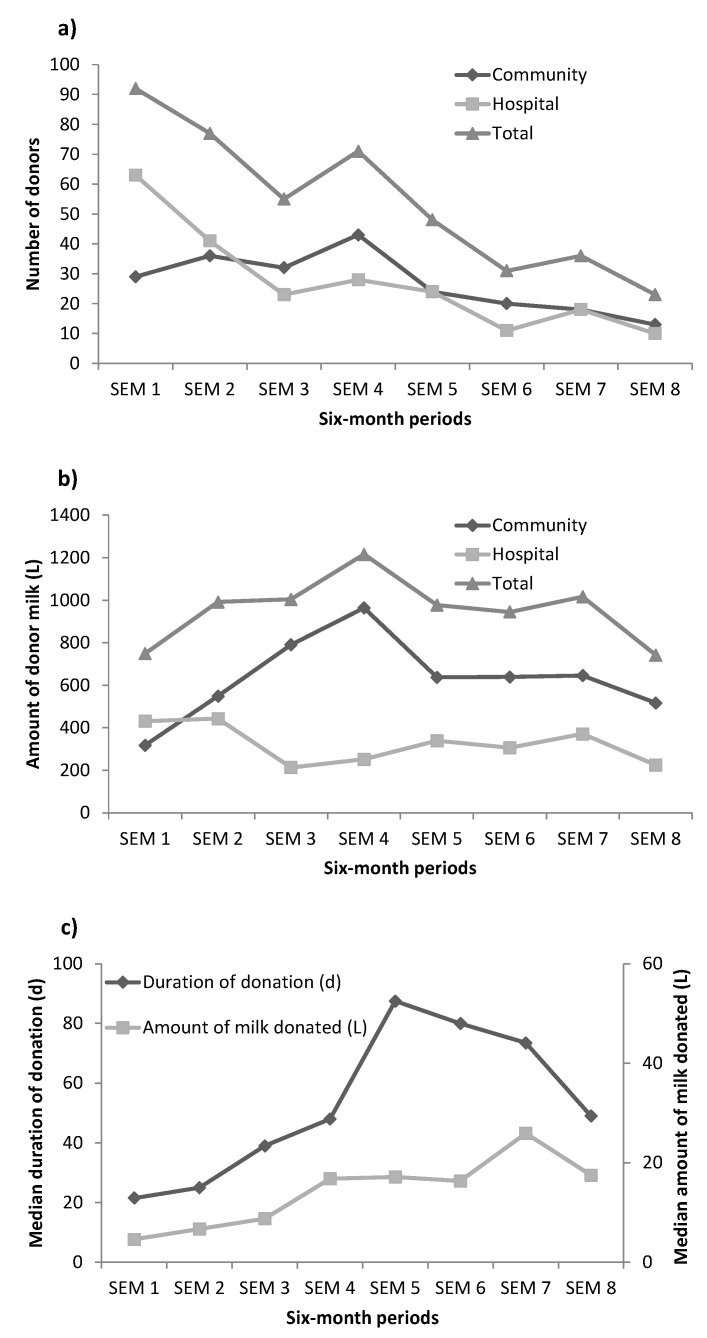
Trend of raw milk collection: number of donors (**a**), amount of donor milk collected (**b**), median duration of donation among women who had stopped donating milk (**c**). SEM, semester–every six months (starting February 2017 and end January 2021).

**Figure 2 nutrients-13-01107-f002:**
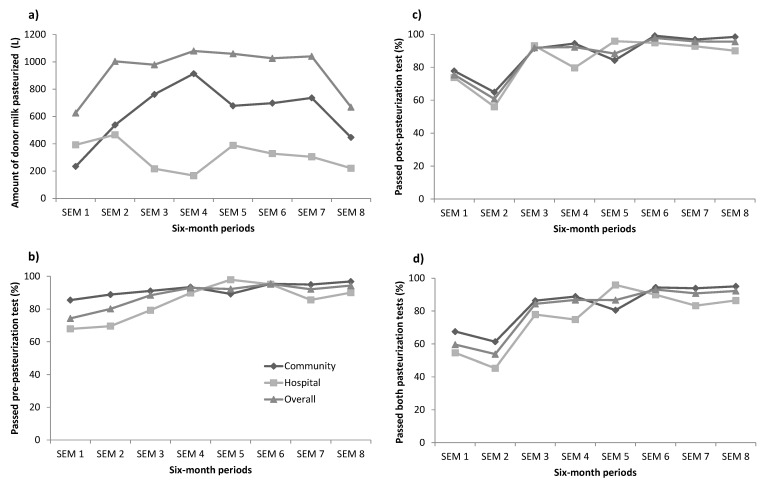
Trend relating to milk handling and processing: amount of donor milk pasteurized (**a**), proportion (%) of milk passed pre-pasteurization test (**b**), proportion (%) of milk passed post-pasteurization test (**c**), proportion (%) of milk passed both pre- and post-pasteurization tests (**d**). SEM, semester–every six months; starting February 2017 and end January 2021.

**Figure 3 nutrients-13-01107-f003:**
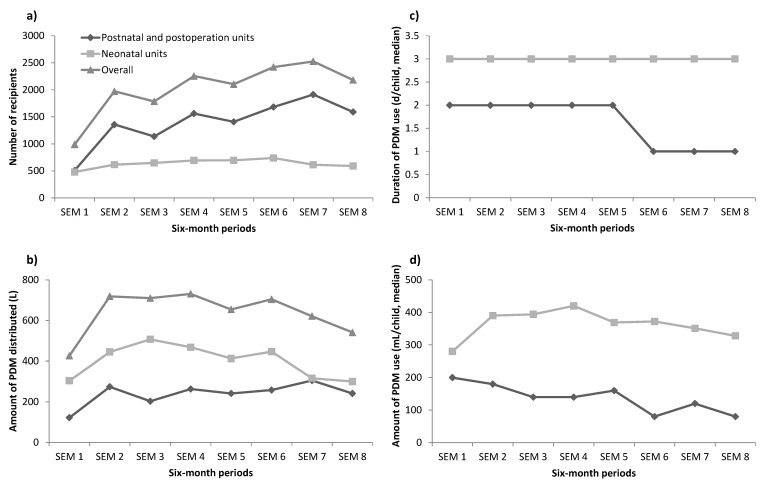
Trend relating to the use of pasteurized donor milk (PDM): number of recipients (**a**), amount of PDM distributed (**b**), median duration of PDM usage (**c**), and median amount of PDM usage (**d**). SEM, semester–every six months; starting February 2017 and end January 2021.

**Table 1 nutrients-13-01107-t001:** Characteristics of human milk donors at enrolment.

	Proportion (%) (*n* = 433)		Proportion (%) (*n* = 433)
Residence Place:		Mean age (SD), y	28.4 (4.1)
Da Nang city	69.8	Number of Children:	
Other provinces	30.2	1 child	58.9
Education:		2 children	37.0
Up to high school	26.1	>2 children	4.1
Diploma	19.6	Birthing place:	
College, university	52.4	DNHWC	79.0
Postgraduate	1.9	Other hospitals in Da Nang	17.8
Profession:		Hospitals from other provinces	3.2
Worker	6.2	Childbirth characteristics:	
Farmer	1.6	Cesarean section	57.0
Housewife	17.6	Preterm birth	43.0
White collar job	74.6	Donors recruited in DNHWC	50.3

Note: DNHWC, Da Nang Hospital for Women and Children.

**Table 2 nutrients-13-01107-t002:** Volume of donor milk collected, pasteurized and passed microbiology tests.

	Community	Hospital	Overall
Amount of donor milk collected, L	5062	2580	7642
Amount of donor milk pasteurized:			
Volume, L	5007	2485	7484
Passed pre-pasteurization test (% of Volume)	92.5	83.1	89.3
Passed post-pasteurization test (% of Volume)	90.0	82.5	87.5
Passed both tests (% of Volume)	85.5	73.7	81.7

**Table 3 nutrients-13-01107-t003:** Characteristics of recipients and amount of pasteurized donor milk (PDM) consumed.

	Neonatal Units (*n* = 5073)	Postnatal Wards (*n* = 11,162)
Characteristics of recipients:		
Preterm (%)	35.8	4.1
Gestation age, week (Mean, SD)	35.2 (3.7)	38.3 (1.4)
Birthweight, g (Mean, SD)	2355 (829)	3141 (496)
Prescription reasons (%):		
Newborns with a birthweight of < 1500 g	15.1	0
Cesarean births	62.8	74.1
Mother absence	4.4	0.2
Mother used medication contradicted to breastfeeding	0.2	0.2
Mothers did not have enough milk	87.9	95.3
Missing data	7.5	4.4
Consumption of PDM:		
Volume of PDM distributed to newborns, L	3199	1907
Average days using PDM, d (median, 25th–75th)	3 (2–4)	2 (1–2)
Average amount of PDM consumed, mL (median, 25th–75th)	370 (198–638)	150 (80–200)

## Data Availability

Requests for data may be directed to the corresponding author and are subject to institutional data use agreements.

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
