# Peer review of "Trends and Dynamics in the First Four Years of Operation of the First Human Milk Bank in Vietnam"

_nutrients, 2021, doi:10.3390/nu13041107_

Round 1
Reviewer 1 Report
Tran et al. performed a study about the characteristics of donors, donation, pasteurization, and recipients of donated human milk during the first four years of operation in the first human milk bank in Vietnam.
I do believe that the paper by Tran et al. was methodologically clear and well written; despite that I have confidence in the fact that the authors should reconsider the strength of their study. Other than that, it will benefit from a few minor revisions.
Minor concerns:
- Line 223 and 229: the authors should include some data elaborated previously to this study in order to clarify these concepts. Ie. Why the preterm milk is better for preterm babies? Milk composition, allergenicity and possible microorganisms. What are the high standard of hygiene that the authors speculate to reach this way?
- Line 258: the authors should include the percentage of increase rate of the donations in the four years they refer to.
Reviewer 2 Report
The manuscript by Hoang Thi Tran et al. is an interesting report, which presents the first four years of operation of the first human milk bank in Vietnam. The subject is interesting but several concerns might help improve the manuscript.
- Please give full name for PDM in Abstract (Line 33)
- Please add a list of abbreviations
- The Introduction section should be expanded. Pleas add the paragraph that will be describe the value of human milk for newborns and infants (i.e reducing the risk of necrotizing enterocolitis, the protection of epithelial cells of newborns and infants against invasion by pathogens)
- Please add to the Methods section, the paragraph, which should be included the outlines of the search strategies, key terms and the criteria of inclusion and exclusion.
- The values, which are mentioned in the text of manuscript (Lines 113 and 147), should be the same as in the Table 1 and 2.
- To the table 1 and 3 please add the information about the number of milk donors/recipients in the given subgroup (n) in relation to all milk donors/ recipients (N) (i.e (% (n/N)). Please add to the Table 1, the information about age of milk donors.
- In Table 3 please provide information about Mean ± SD for Birth weight and Gestational age. Moreover, in my opinion for the results contained in the table 3, should be performed The chi-square test.
- It would be interesting to list the bacteria present in the milk pre- and post-pasteurization. Were there any differences in the presence of pathogens depending on where the milk was collected (with exception Bacillus bacteria, which are mentioned in the Discussion section)
- The names of the bacteria should be in italics (i.e.Bacillus)
However, I believe that the work is overall valid.
Reviewer 3 Report
The submitted to Nutrients manuscript entitled “Trends and dynamics in the first four years of operation of the 2 first human milk bank in Vietnam" by Hoang Thi Tran and al. contains interesting and actual results.
It provides interesting information on the functioning of human milk banks in Vietnam. These milk is especially important for premature babies.
The research methodology does not raise any objections.
Below, I would like to note some questions concerning the details that I hope will be useful for the authors:
- What, according to the authors, was the reason for the decrease in the amount of milk donated to the bank, despite the increase in awareness and education of women?
- There is no information at work about the exact tests performed to ensure the high quality and safety of the collected milk for future consumers .
- Have different microbial and chemical contaminants been taken into account?Please describe.
- The presented statistical analysis of the collected data is quite poor, please complete.
Authors should carefully check for grammar, punctuation and sentence structure before submitting the revised paper. Literature citation should be in accordance with the requirements of the publisher. Check the whole manuscript, please.
